# Mechanical and Microstructural Characterization of an AZ91–Activated Carbon Syntactic Foam

**DOI:** 10.3390/ma12010003

**Published:** 2018-12-20

**Authors:** Nima Movahedi, Mehdi Taherishargh, Irina V. Belova, Graeme E. Murch, Thomas Fiedler

**Affiliations:** Centre for Mass and Thermal Transport in Engineering Materials, School of Engineering, The University of Newcastle, Callaghan, NSW 2308, Australia; nima.movahedi@uon.edu.au (N.M.); mehdi.taherishargh@newcastle.edu.au (M.T.); irina.belova@newcastle.edu.au (I.V.B.); graeme.murch@newcastle.edu.au (G.E.M.)

**Keywords:** foam, magnesium, deformation, casting, compression, properties, syntactic, density

## Abstract

In this study, activated carbon (AC) particles were combined with AZ91 alloy to manufacture a magnesium syntactic foam. This novel lightweight foam has a very low density, in the range of 1.12–1.18 gcm^−3^. The results show that no chemical reaction occurred between the AZ91 matrix and the activated carbon particles. The mechanical properties of the foam were evaluated under quasi-static compression loading conditions, and showed a consistent trend for the energy absorption of the fabricated AZ91–AC syntactic foams. The deformation mechanism of samples was a brittle fracture mode with the formation of shear bands during the fracture of all samples.

## 1. Introduction

On account of their low density and sound and energy absorption characteristics, porous materials have recently attracted a significant amount of attention. These materials demonstrate unique physical and mechanical properties, making them especially suitable for use in various industries, such as the construction, automotive and medicine industries [1]. Metallic foams are a subset of porous materials. Because of their useful mechanical properties, they have been widely produced and evaluated [2]. The various types of metallic foams available have been manufactured with different techniques [3,4,5,6]. Magnesium, due to its high biocompatibility and lower density than other types of metallic materials, could be one of the most interesting and suitable candidates for the production of metallic foams [7]. Different types of Mg alloy foams (with closed or open cell structures) have been manufactured with different methods, such as casting or powder metallurgy [8,9]. An important subgroup of metal foams is metal matrix syntactic foam, which is composed of porous or hollow particles that are embedded in a metallic matrix [10]. In recent years, research has mostly focused on the production and investigation of aluminum syntactic foams [11,12,13]. Magnesium alloys have been considered for foam production due to their high specific strength. The few published studies have used cenosphere and fly ash particles as fillers [14,15,16] to produce the syntactic foams. However, these fillers are susceptible to fracture and melt infiltration, which is amplified by a potential chemical reaction with the molten metal during casting. To decrease the proportion of fractured particles, cenospheres have been surface-coated to prevent their direct contact with molten magnesium [17]. However, this additional surface treatment increases the complexity and manufacturing costs. Even so, a surface treatment may add additional benefits as it can be used to optimize the essential interphase between the particles and the matrix. Moreover, due to the high chemical reactivity of the metal, corrosion resistance is an important factor that must be considered for the application of Mg foams. According to [18,19], the corrosion resistance of Mg syntactic foam was enhanced in comparison to pure Mg due to the incorporation of filler particles. This was explained by observing changes in the Mg’s microstructure and a decrease in the exposed Mg surface area due to partial coverage by inert silica particles. Conversely, particles that are permeable to a corrosive medium are likely to increase the corrosion rate due to the large internal surface area of a foam.

The aim of the present research work is to manufacture an innovative magnesium syntactic foam (MSF) using chemically inert filler particles. At ambient pressure, magnesium and carbon do not produce stable compounds at any temperature [20]. Therefore, activated carbon (AC) with an ~2.8 mm particle size was used as a filler within an AZ91 magnesium alloy matrix. Micron-sized hollow AC particles have previously been used to produce a pure open-cell magnesium foam for bio-applications [21]. The micron-sized activated carbon particles in [21] were hollow spherical types that were removed after casting to produce an open-cell Mg foam. In the present research, granular and porous activated carbon particles were used instead and remained in the material to produce the magnesium syntactic foam. The research study [21] focused predominantly on the oxidation and corrosion behavior of Mg foams. In contrast, the present study addresses microstructural analysis and mechanical properties evaluation. In this study, six samples of a novel AZ91/AC syntactic foam were successfully manufactured for the first time and subjected to microstructural (using a scanning electron microscope (SEM) and energy dispersive spectroscopy (EDS)) and mechanical testing. Due to the high specific strength of the magnesium AZ91 alloy, this material may be of interest for structural applications.

## 2. Materials and Methods

Activated carbon (AC) is an inexpensive carbonaceous material with a low bulk density (0.47 gcm^−3^), a high specific surface area, and excellent chemical stability. It is widely used for water treatment to absorb impurities from drinking water. The term ”activated” refers to the final step of AC production to achieve a porous carbonaceous structure [22]. Activated carbon particles were obtained from Activated Carbon Technologies Pty in Australia. AZ91 magnesium alloy (9 wt.% Al, 1 wt.% Zn, 0.5 wt.% Mn, balance Mg) was used as the matrix of the metal syntactic foam. AZ91/AC syntactic foams were produced using an established counter-gravity infiltration technique [11,23]. For this purpose, a graphite mold was filled in four individual packing and vibration steps with equal amounts of activated carbon particles in each step. This compaction method minimizes the density gradient throughout the sample height. After this step, a stainless steel mesh was inserted into the open end of the mold to keep the activated carbon particles in place. The assembly was performed inside an argon-filled glovebox to minimize the oxygen level in the system. To this end, the AZ91 alloy block was inserted inside the graphite crucible and the filled graphite mold was inserted upside-down inside the crucible. A steel cap was then put on top of the mold to collect excess melt during infiltration casting. In the next step, these parts were located inside the stainless steel sleeve. Finally, the assembly was placed inside an electric furnace and kept for 45 min at a temperature of 660 °C while a protective argon atmosphere was used to prevent oxidative combustion of the magnesium alloy during the casting process. The metallic melt was forced to infiltrate a packed bed of activated carbon particles by applying a 0.02 MPa pressure. After cooling the mold in atmospheric conditions, the produced sample was manually removed from the mold. Figure 1 shows a schematic of the utilized setup.

Figure 2a shows an AZ91–AC syntactic foam sample. All samples were cylindrical with a diameter of 28.83 ± 0.02 mm and a height of 42.33 ± 0.19 mm. The density of the cylindrical foam samples was calculated by dividing their total mass by their macroscopic cylindrical volumes. The volume fraction of AZ91 matrix ($F_{\mathrm{AZ}91})$, activated carbon filler ($F_{\mathrm{AC}})$, and voids $\left(F_{V}\right)$ within the structure of the produced syntactic foams were calculated according to the equations below [23]:(1)$$F_{\mathrm{AZ}91}=\frac{\frac{m_{\mathrm{SF}}-m_{\mathrm{AC}}}{\rho_{\mathrm{AZ}91}}}{V_{\mathrm{SF}}}$$
(2)$$m_{\mathrm{AC}}=\rho_{B}\times V_{\mathrm{SF}}$$
(3)$$F_{\mathrm{AC}}=\frac{\rho_{B}}{\rho_{\mathrm{AC}}}$$
(4)$$F_{V}=1-F_{\mathrm{AZ}91}-F_{\mathrm{AC}}$$
where $m_{\mathrm{SF}}\;\mathrm{and}\;m_{\mathrm{AC}}$ are the masses of the produced syntactic foam and activated carbon, respectively, $V_{\mathrm{SF}}$ is the syntactic foam volume, and $\rho_{\mathrm{AZ}91}$ is the solid density of the AZ91 matrix alloy (1.81 gcm^−3^). The densities $\rho_{B}$ and $\rho_{\mathrm{AC}}$ are the bulk and envelope density of activated carbon particles and are 0.47 and 0.8 gcm^−3^, respectively.

The microstructure study of the produced syntactic foams was performed using a scanning electron microscope (Zeiss Sigma VP FE, Zaventem, Belgium) equipped with a Bruker EDS (Billerica, MA, USA). SEM analysis was first used to analyze the interface between matrix and fillers. To this end, one sample was cut longitudinally near its central axis using a lathe machine. In order to avoid damage or contamination of the AZ91 matrix and filler, this first SEM study was performed on the unpolished surface. In addition, X-ray diffraction (Philips X’Pert MPD, Amsterdam, The Netherlands) was performed to identify the chemical composition of the AZ91/AC syntactic foam. A second SEM analysis addressed the microstructure of the AZ91 struts and, hence, required a polished surface (see Figure 2b). To this end, the opposite cross-sectional area (from the other sample half) was manually polished using silicon carbide grinding paper with grit sizes of 180, 240, 320, 600 and 1200. The final polishing was performed using emery paper and a diamond suspension to attain a mirror-like surface (Buehler, IL, USA).

Quasi-static compressive tests were performed on a SHIMADZU 50 KN uni-axial testing machine (Kyoto, Japan) at a crosshead speed of 1 mm/min. Both ends of the cylindrical samples were lubricated before the compression tests to minimize friction between the sample surfaces and compression platens. The force-displacement data were then converted to engineering stress-strain curves according to the initial dimensions of the samples. The main mechanical properties of the samples were evaluated according to ISO 13314-11 [24]. In the first step, the sample with the lowest density was compressed and its plateau stress was determined to define the loading–unloading cycle for the remaining samples. The unloading cycle is used to eliminate the settling effect of samples and defines a second elastic modulus in addition to the quasi-elastic modulus, which is calculated from the first slope of the stress-strain curve. The 1% proof stress was determined using the quasi-elastic modulus (i.e., the initial gradient of the stress-strain curve). The energy absorption and energy absorption efficiency of the AZ91/AC syntactic foams were calculated using the following equations [24]:(5)$$W=\int_{0}^{0.5}\sigma \;d\epsilon$$
(6)$$\eta =\frac{W}{0.5\times \sigma_{\max}}$$

## 3. Results and Discussion

Table 1 shows the physical properties of the manufactured AZ91–AC syntactic foam samples. The densities of the samples vary between 1.12 and 1.18 gcm^−3^. The void fraction of the sample decreases with density. The average density of the novel AZ91/AC syntactic foam is 1.14 gcm^−3^ and is close to the density of some of the open-cell magnesium foams reported in the literature with 1.13 and 1.03 gcm^−3^ [25]. The density of the samples in this research study is lower compared to AZ91 syntactic foams with hollow glass fillers (1.4–1.6 gcm^−3^) [14]. The produced AZ91–hollow alumina foam samples in [26] also show a considerably higher density in comparison to the AZ91–AC syntactic foams. A likely explanation is the higher bulk density of hollow alumina particles compared to the AC fillers. Moreover, the filling of some damaged hollow alumina fillers with molten magnesium could have increased the density of the AZ91–hollow alumina syntactic foam samples in [26]. Compared to [27], which presents an MSF with hollow SiC particle filler, the density of the samples produced in this research is about 15% higher.

Figure 2a shows an example of the produced AZ91/AC syntactic foam. The microstructure of an as-cast AZ91 strut is shown in Figure 2b and is composed of different phases [28]. The EDS analysis showed that it contains an α- matrix (95.14 at.% Mg, 4.64 at.% Al, 0.21 at.% Zn and 0.01 at.% Mn), a β-Mg_17_Al_12_ phase (71.53 at.% Mg, 27.08 at.% Al, 1.37 at.% Zn and 0.02 at.% Mn) and a eutectic α + β phase. Some manganese-rich areas additionally contain Al–Mn phases. A backscattered SEM image of the AZ91 matrix and the activated carbon is shown in Figure 3a. Unlike the fillers mentioned in other studies, the activated carbon particles did not strongly react with the magnesium alloy during production. The AC particles remained intact with no visible mechanical damage. Some signs of reaction layers at the interface between the filler and the matrix were detected (see Figure 3a). An EDS line scan (Figure 3b) across this reaction layer indicates the presence of magnesium and oxygen that can most likely be attributed to the surface oxidation of the struts. Due to its high specific surface area, activated carbon is prone to absorb oxygen from air at room temperature and release it at elevated temperatures during casting [29]. This may explain the partial matrix oxidation despite the presence of a protective argon atmosphere. It should be highlighted that no carbon was detected within this reaction layer. XRD analysis of an AZ91–AC syntactic foam sample shows the diffraction peaks of MgO, which further verifies the results of the EDS analysis. No magnesium carbides were detected and, hence, possible reaction between the matrix and filler particles can be excluded (Figure 3c).

The compressive stress-strain curves of the AZ91/AC syntactic foam are shown in Figure 4a. Their shape is quite characteristic for a metallic foam, and exhibits the consecutive pseudo-elastic, plateau and densification regions. All stress-strain curves show a gradual stress increase in the pseudo-elastic region up to a peak value followed by a stress drop because of shear band formation (see Figure 4a). A similar transition from a peak stress towards a plateau was reported for AZ91/hollow SiC syntactic foams [27]. At high strains, densification of the AZ91/AC foam occurred and is visible by a sharp stress increase. Due to the utilized manufacturing procedure, the volume fraction of activated carbon particles is considered to be constant in all of the produced syntactic foams. The deformation mechanisms of all samples are also identical. Selected mechanical properties of the syntactic foams according to ISO 13314 are shown in Table 2. The peak stress varies from 32.45 to 51.43 MPa for AZ91–AC syntactic foams. The observed compressive strength of the samples in this study is higher compared to AZ91–hollow SiC syntactic foam (about 27 Mpa) [27]. The average plateau stress of the AZ91/AC syntactic foams in this study is about 12.71 MPa. This value is close to the plateau stress of the AZ91/hollow SiC syntactic foam in [27], and considerably higher than the plateau stress of an AZ91–microballoon foam where its compressive stress-strain curve does not show any remarkable plateau and densification regions and almost behaves like a ceramic foam [30]. The plateau stress and energy absorption of foam materials are dependent on the sample density [1]. As can be seen, the volumetric energy absorption generally increases with density. The macroscopic deformation of an AZ91–AC syntactic foam sample (density 1.15 gcm^−3^) is presented in Figure 4b. The compressive stress of this sample reaches its maximum value at ϵ=0.08. The photograph of the sample at this strain shows some signs of barreling and the formation of small cracks. With increasing compressive strain, existing cracks grow whilst new ones emerge. Most cracks form an angle of about 45° relative to the compressive load, which is indicative of shear failure. Brittle fracture along these shear bands decreases the strength of the syntactic foam and results in the sudden stress drop visible in the stress-strain data (see Figure 4a, ϵ=0.16). The brittle deformation of the AZ91–AC syntactic foam samples continues until the onset of densification with multiple serrations. In AZ91 alloys, brittle fracture is the main deformation mode. This behavior is mainly attributed to the presence of the brittle β-Mg_17_Al_12_ phase in the as-cast AZ91 alloys [28].

## 4. Conclusions

A lightweight AZ91–AC syntactic foam was produced with an average density of 1.14 gcm^−3^, which is below the previously reported densities for magnesium syntactic foams with fly-ash or microsphere fillers. Microstructural studies on the interface of the AZ91 Mg matrix and activated carbon fillers confirmed no evidence of any severe chemical reaction. The deformation behavior of the produced AZ91–AC syntactic foams revealed a brittle deformation behavior along with some signs of barreling under quasi-static compression. The measured mechanical properties parameters showed consistency among the produced syntactic foams. However, any variation in some mechanical properties can largely be attributed to the deformation mechanism of the foams.

## Figures and Tables

**Figure 1 materials-12-00003-f001:**
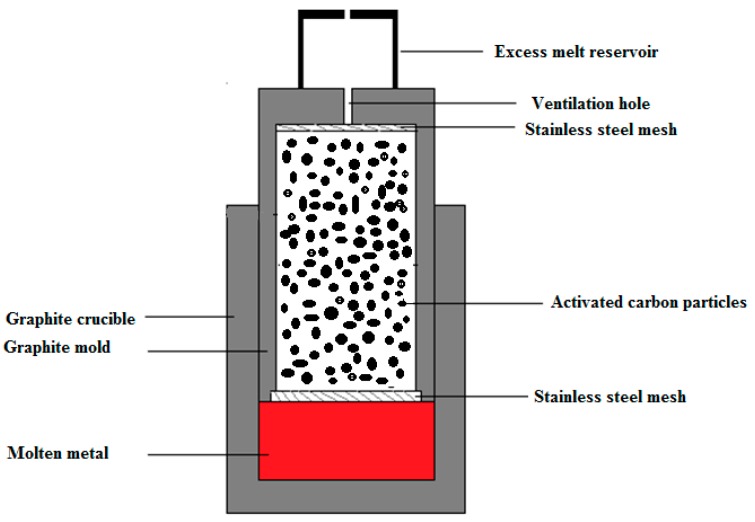
A schematic of the counter gravity infiltration casting technique.

**Figure 2 materials-12-00003-f002:**
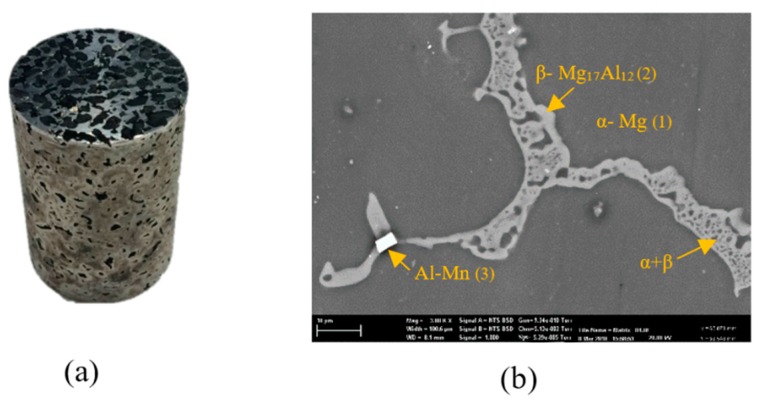
(**a**) The macrostructure of the AZ91–activated carbon (AC) syntactic foam, (**b**) the microstructure of an as-cast strut.

**Figure 3 materials-12-00003-f003:**
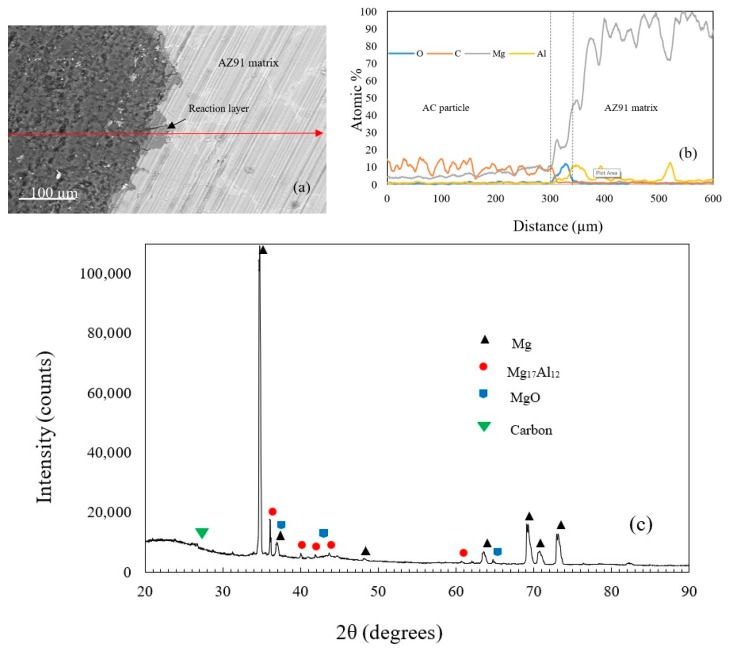
(**a**) SEM images of the interface between the matrix and an AC particle, (**b**) an EDS line scan, (**c**) the XRD pattern of the AZ91–AC syntactic foam.

**Figure 4 materials-12-00003-f004:**
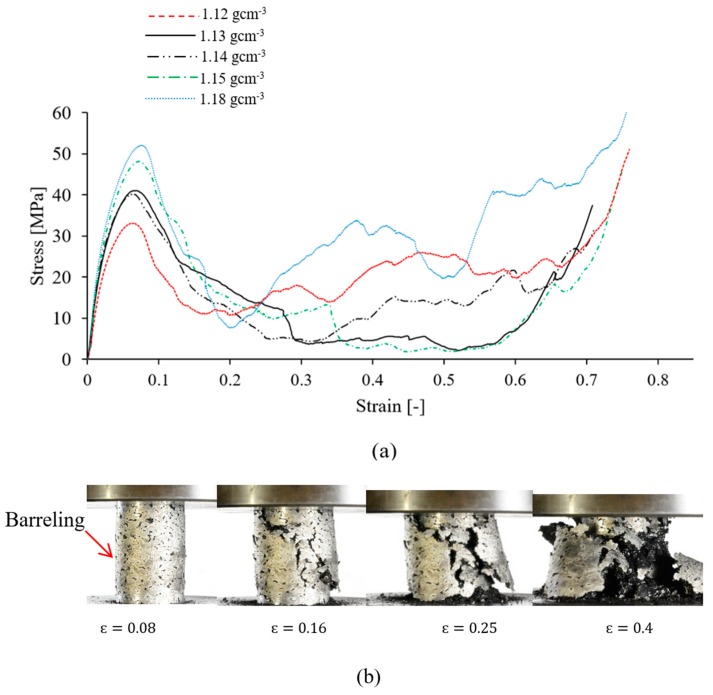
(**a**) Compressive stress-strain curves of AZ91–AC syntactic foams with different densities, and (**b**) macroscopic deformation of an AZ91/AC syntactic foam with a density of 1.15 gcm^−3^.

**Table 1 materials-12-00003-t001:** The physical properties of the AZ91–AC syntactic foams.

Sample No	SF Density (gcm^−3^)	*F*_AZ91_ (%)	*F*_AC_ (%)	*F*_v_ (%)
MSF_1_	1.12	36.01	58.75	5.24
MSF_2_	1.13	36.51	58.75	4.74
MSF_3_	1.14	36.92	58.75	4.33
MSF_4_	1.15	37.32	58.75	3.93
MSF_5_	1.18	39.38	58.75	1.87

MSF, magnesium syntactic foam.

**Table 2 materials-12-00003-t002:** Selected Mechanical Properties of the AZ91–AC syntactic foams.

Sample Density (gcm^−3^)	Quasi-Elastic Gradient (MPa)	Peak Stress (MPa)	Plateau Stress (MPa)	Energy Absorption (MJ/m^3^)
1.12	1467.8	32.45	16.01	9.53
1.13	1663.34	40.90	8.25	7.61
1.14	1890.00	39.30	7.03	7.85
1.15	1889.96	48.00	9.52	8.30
1.18	2667.82	51.43	22.74	13.49

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
