# Peer review of "Mechanical and Microstructural Characterization of an AZ91–Activated Carbon Syntactic Foam"

_materials, 2018, doi:10.3390/ma12010003_

Reviewer 1 Report

In the first paragraph of the introduction part, authors mentioned that the surface treatment of the fillers increases the complexity and manufacturing expenditure. For metal matrix composites, the interface between reinforcement and matrix is the very essential factor determined the properties of the composites. The description about the surface treatment herein maybe not suitable for the explanation of the advantages of the current research.

Authors cited reference 19 to show micro-sized AC particles have been previous used to produce a pure open-cell magnesium foam for bio-applications. What is the relationship between the reference and current research. And there is no illustration in main text to show what is the advantages and disadvantages of the current research compared to reference 19.

In the last paragraph of the introduction part, authors mentioned “due to the increased strength of the magnesium AZ91 alloy, this material may be of interest for structural applications.” It is a really good thing that the research is application driven and contributes the development of

Mg in real application. When considering the structural application, the authors should supply the corrosion behavior of the Mg foams in terms of the introduction of the foreign particles into matrix will worsen the corrosion of Mg. 

Author Response

We want to thank the referee for his valuable remarks. Please find a detailled answer letter in the attached document. 

Reviewer 2 Report

Last years show increase of unchangeable interest in porous materials that are most absorbing research topic. Mechanical characterization is mostly investigated subject among others. Presented article is of high scientific quality and sufficiently describes interesting and important problem of mechanical application, especially from energy dissipation or noise suspension theme point of view.

Answers for below mentioned questions and suggestions would make the paper ready for acceptance:

1.     If the Authors decided to add a gravity infiltration technique scheme, this would add clarity to the description of the method.

2.     Do the Authors of the study used SEM sample in original size, and conducted surface research, or whether they were to be conducted on the cross section. If it was tested on the cross section, how did the authors conduct a metallographic preparation of such a delicate system, without damaging structure.

3.     Was mold coolled by natural or accelerated proccess?

Author Response

Thank you for the helpful inputs to our manuscript. Please find our replies attached.

Reviewer 3 Report

The present paper deals with the fabrication of magnesium syntactic foam by the use of activated carbon particles combined with AZ91 alloy.

The reviewer ask to clarify the following points:

- In the introduction section please define "MSF" and "EDS".

- The reviewer suggests to change the section order of section 2 "Results and discussion" and section 3 "Materials and Methods".

After this minor revision, I suggest the present article for publication

Author Response

Thank you for this positive assesment. Please find a description of our modifications attached.

Reviewer 4 Report

In this manuscript, the authors studies the microstructure and compressive mechanical properties of AZ91 matrix filled activated carbon (AC) particles synthesized by infiltration method. The reviewer doesn't have any comment about the result but really doubt about the methodology of this work:

How many specimens were used to test? It seems there are only 3 specimens with the density of 1.13, 1.14, and 1.15 kg/m3. If yes, it isn’t enough to provide the fair conclusion of the material properties.

Although the title of the manuscript is studying the synthesis method of AZ91/AC composite materials, the authors only studied the microstructure and mechanical properties of composite materials without considering the effect of parameters that were setup for the fabrication method.

Is it correct when using the term "syntactic foam" here? The definition of syntactic foam is a foam material having the porous structure by filling the hollow particles. In this research, the void fraction of these material is very low (around 4 vol.%), the density of these decreases because of filling high volume percentage of low density AC particles. The reviewer thinks that using the term of composite or light weight composite material is more accurate for these.

Other comments:

Mentioning about using XRD and SEM to study the microstructure is unnecessary in the abstract.

What is the authors’ purpose of showing the Fig.2(d)? The pore is not captured properly in that image. 

Figure 3.a: The line type of the stress-strain curve needs modification because it is difficult to recognize in the print-out version. The axes plot without stick mark.

The photograph of the sample at this strain shows some signs of barreling and the formation of small cracks”. Actually, these signs can be seen in the first image of Figure3.b. Maybe, the image is too small.

The authors didn't give any discussion about the role of particles in the fracture condition.

What is the quasi-static gradient?

Finally, the reviewer requires a major revision for this manuscript.

Author Response

Answer Letter to Reviewer IV

In this manuscript, the authors studies the microstructure and compressive mechanical properties of AZ91 matrix filled activated carbon (AC) particles synthesized by infiltration method. The reviewer doesn't have any comment about the result but really doubt about the methodology of this work:

How many specimens were used to test? It seems there are only 3 specimens with the density of 1.13, 1.14, and 1.15 kg/m3. If yes, it isn’t enough to provide the fair conclusion of the material properties.

This is a valid criticism. Two more samples were manufactured and tested with the densities of 1.12 and 1.18 gcm-3 and the manuscript was revised accordingly.

Although the title of the manuscript is studying the synthesis method of AZ91/AC composite materials, the authors only studied the microstructure and mechanical properties of composite materials without considering the effect of parameters that were setup for the fabrication method.

Following the suggestion of the referee, the title of the manuscript has been changed to “Mechanical and microstructural characterization of AZ91-Activated Carbon Syntactic Foam”

Is it correct when using the term "syntactic foam" here? The definition of syntactic foam is a foam material having the porous structure by filling the hollow particles. In this research, the void fraction of these material is very low (around 4 vol.%), the density of these decreases because of filling high volume percentage of low density AC particles. The reviewer thinks that using the term of composite or light weight composite material is more accurate for these.

Due to the porous nature of activated carbon (albeit a low porosity as the reviewer points out correctly) we believe that the term syntactic foam is still appropriate.

Other comments:

Mentioning about using XRD and SEM to study the microstructure is unnecessary in the abstract.

The corresponding sentence has been removed.

What is the authors’ purpose of showing the Fig.2(d)? The pore is not captured properly in that image. 

The mentioned image has been deleted.

Figure 3.a: The line type of the stress-strain curve needs modification because it is difficult to recognize in the print-out version. The axes plot without stick mark.

The line type of stress-strain curve was modified for better visualization. The axes were plotted with tick marks (please see Fig.4a)

 “The photograph of the sample at this strain shows some signs of barreling and the formation of small cracks”. Actually, these signs can be seen in the first image of Figure3.b. Maybe, the image is too small.

The image size and the text have been adjusted as suggested (please see Fig. 4b).

The authors didn't give any discussion about the role of particles in the fracture condition.

In the opinion of the author’s the available data does not support such a discussion as only the properties of the composite were determined.

No study of a related foam material with removed or different particles was found in the literature. Hence, a comparative analysis and discussion was unfortunately not possible.

What is the quasi-static gradient?

We could not find any mention of a quasi-static gradient. We assume the referee refers to the quasi-elastic gradient. The following information has been added to the text: “quasi-elastic modulus (i.e. the initial gradient of the stress-strain curve).”

Finally, the reviewer requires a major revision for this manuscript.

We hope we have been able to address the reviewer’s concerns.

Round  2

Reviewer 4 Report

In general, I agreed with the authors' reply and the manuscript is recommended to publish in the current form. Congratulations!